# Potential Role of Non-Steroidal Anti-Inflammatory Drugs in Colorectal Cancer Chemoprevention for Inflammatory Bowel Disease: An Umbrella Review

**DOI:** 10.3390/cancers15041102

**Published:** 2023-02-09

**Authors:** Peri Newman, Joshua Muscat

**Affiliations:** Department of Public Health Sciences, Penn State College of Medicine, Hershey, PA 17033, USA

**Keywords:** inflammatory bowel disease, colorectal cancer, chemoprevention, NSAIDs

## Abstract

**Simple Summary:**

Non-steroidal anti-inflammatory drugs prevent the recurrence of adenomatous polyps and colon cancer. At low doses, with presumed low gastrotoxicity, there has been substantial interest in NSAIDS as cancer prevention agents. Their potential in high risk groups, including inflammatory bowel disease, may be particularly beneficial. In this review, we summarize the medical opinions on this topic and suggest areas of future research.

**Abstract:**

Inflammatory Bowel Disease (IBD) is a category of autoimmune diseases that targets the destruction of the gastrointestinal system and includes both Crohn’s Disease and Ulcerative Colitis (UC). Patients with IBD are at a higher risk of developing colorectal cancer (CRC) throughout their lives due to chronically increased inflammation. Nonsteroidal anti-inflammatory drugs (NSAIDs) are potential chemopreventative agents that can inhibit the development of CRC in persons without IBD. However, the use of NSAIDs for CRC chemoprevention in IBD patients is further complicated by NSAIDs’ induction of damage to the bowel mucosal layer and ulcer formation. There has been a push in new research on chemopreventative properties of certain NSAIDs for IBD. The purpose of this umbrella review is to investigate the potential of low-dose NSAID compounds as chemopreventative agents for patients with IBD. This paper will also suggest future areas of research in the prevention of CRC for patients with IBD.

## 1. Introduction

Inflammatory bowel disease (IBD) is a category of autoimmune-related gastrointestinal diseases in which the intestinal tract is targeted for injury by an over-active immune system. There are two common types of IBD, Crohn’s Disease and Ulcerative Colitis, while a third and lesser-known type, microscopic colitis, is included [1]. The age demographic for an IBD diagnosis is bimodal, with a peak in diagnosis in people under the age of 30 and another peak for people over 60 years old [2]. Crohn’s Disease is characterized by lesions and inflammation that can occur in any area of the gastrointestinal tract [3]. Ulcerative Colitis (UC) is known to be manifested by ulcers present in the colon, and microscopic colitis is characterized as a lack of endoscopic abnormalities and a pathological influx of immune cells and mucosa damage [4,5]. For all three types of IBD, chronic, watery, diarrhea and abdominal pain encompass the common symptoms. The differentiation of the type of IBD is conducted via endoscopic and pathologic methods. In addition to an autoimmune response, both environmental and genetic factors have been hypothesized to increase one’s risk and onset of IBD. IBD is treated with steroids, non-steroidal anti-inflammatory drugs (NSAIDs), biologics, and, in severe cases, surgery to remove the damage [6,7]. The incidence of IBD has increased worldwide within the last few decades [8]. Within North America, the pooled prevalence of IBD is 0.3% in the 21st Century and is expected to increase past 0.6% by 2030 [9]. In Westernized nations, IBD is more common in Caucasians and those of Ashkenazi descent. In adults, the incidence of Crohn’s Disease is higher in women. However, developing nations are experiencing increasing rates of IBD [8]. Asian countries are experiencing prevalence increases. Taiwan has experienced a prevalence increase in CD from 0.6 to 2.1 per 100,000 people and a prevalence increase in UC from 3.9 to 12.8 per 100,000 people [9]. In China, IBD prevalence has increased from approximately 12,000 people to 266,394 people from 2000 to 2010 [8]. The reasons for these changes are not well understood, partly reflecting questions on the quality of the data [10]. One factor of IBD that is not often discussed is the financial burden placed on patients and their caretakers. The direct annual U.S. healthcare costs for the treatment of IBD are estimated to be well into the billions [11]. The cost of IBD spreads beyond just finances and lifestyle changes; anemia, psychological distress, and arthritis are among the most cited comorbidities of IBD [12,13,14]. As the prevalence of IBD increases and pushes IBD towards global status, it is important to understand the impacts this disease has on populations. 

One burden of patients with IBD is the potential development of colorectal cancer (CRC). Patients with Crohn’s Disease and Ulcerative Colitis are at an increased risk of developing CRC; however, microscopic colitis has not been found to be associated with an increased risk of CRC. In North America, when compared to the general population, patients with CD are 2.64 times more likely to develop CRC [95% CI: 1.69–4.12] and patients with UC are 2.74 times more likely to develop CRC [95% CI: 1.91–3.97] [15]. The role of traditional risk factors in the development of CRC, such as diet, obesity, physical activity etc., among IBD patients has not been evaluated. The molecular mechanisms are thought to be similar for patients without IBD but the chronic inflammation from IBD facilitates the rapid progression of DNA damage induced by oxidative stress within the gastrointestinal tract [16,17]. As more damage to the intestinal tract occurs, dysplasia polypoid lesions form and can continue to develop into carcinomas [1]. The chemoprevention of CRC among patients with IBD is a topic of great importance, particularly for those who are diagnosed with CRC at a young age. Although the mechanisms of IBD’s progression to CRC are unclear, immunosuppressive therapy for IBD may increase tumor growth and progression; however, there is little data to support or refute this notion. In theory, the decrease in chronic inflammation accounts for the observations of a decreased risk of CRC development in persons taking NSAIDs. Supplementing standard IBD treatments with promising chemopreventative agents has also been hypothesized to decrease the risk of CRC [1,18].

The use of NSAIDs, particularly aspirin, as a chemopreventative agent for CRC has been widely studied and it is agreed that NSAIDs are effective at lowering the risk of CRC and recurrent adenomas [19]. 5-ASA medications are aminosalicylic acids that are chemically related to aspirin and, similarly to NSAIDS, they work by inhibiting COX-related prostaglandins (PGE2), LOX-related leukotrienes and histamines, and cytokines of the inflammatory process, thus allowing damaged tissue from IBD to heal. Technically, 5-ASA agents are not NSAIDs, but they share a structural similarity with aspirin, differing only in the presence of an amino group at position five of the benzene ring. The 5-ASA agents are extremely useful in the treatment of IBD as they are less toxic to the gastrointestinal lining than conventional NSAIDs. The risk estimates vary depending on type of NSAID, dose and duration, but range between approximately 25–35% reduced risk. Effects have been shown for even 1 year or less of usage. Cyclooxygenase (COX) is an enzyme that converts arachidonic acid into prostaglandins and is known to have two isoforms: COX-1 and COX-2 [20]. The COX-2 pathway is known to contribute to tumorigenesis through the downstream signaling of prostaglandin, leading to upregulated cell proliferation. NSAIDs target COX-2, leading to potential downregulation of cancerous cell growth [21]. NSAIDs have been shown to attenuate downstream Nf-ĸB -mediated DNA transcription, leading to suppressed cell growth [22]. Additionally, NSAIDs inhibit the expression of IL-1B, a known pro-inflammatory cytokine. By reducing the IL-1B levels, an anti-tumor response is initiated by the immune system [23]. Therefore, the hypothesis of NSAIDs as a chemoprotective agent is mechanistically sound. Multiple randomized control studies and large-scale cohort studies have been in support of low-dose aspirin as an effective chemopreventative agent for CRC [24,25,26]. Whereas NSAIDs are used to treat chronic inflammatory conditions, such as arthritis and musculoskeletal disorders in middle-aged to older adults, their use for the abdominal pain treatment of IBD is intermittent. It is recommended that NSAIDS should not be taken during periods of flare-ups. However, one study found that recurrent Crohn’s disease can be triggered by occasional or frequent NSAID use in some patients [27]. Furthermore, aspirin and other NSAIDs have been shown to exacerbate symptoms in patients with IBD [28,29]. It may be that NSAIDs at typical doses for pain relief are contraindicated for Crohn’s Disease, and possibly IBD in general. A few studies have suggested that low-dose aspirin does not contribute to IBD exacerbation, but the dose-response relationship is still largely unclear [30,31,32]. There are gaps in the literature concerning the use of low-dose aspirin and other NSAIDs as chemopreventative agents for the subset of individuals at a markedly higher risk for CRC; namely, patients with IBD. This umbrella review is conducted to summarize the literature and the gaps in the research to investigate the potential of low-dose NSAID compounds as chemopreventative agents for patients with IBD. This paper will also suggest future areas of research in the prevention of CRC for patients with IBD.

## 2. Materials and Methods

The PUBMED and Cochrane Library databases were searched between September 15 and November 2022 to determine the relevant articles. Within the Cochrane Library, “‘gastroenterology and hepatology’ and ‘cancer: colorectal’” and “‘gastroenterology and hepatology’ and ‘inflammatory bowel disease’” were the two searches conducted. The following searchers were used in PUBMED:“nsaid*”[All Fields] AND “ibd”[All Fields] AND “cancer*”[All Fields]((nsaid*) AND (colorectal cancer) AND (ibd))((aspirin) AND (colorectal cancer)) AND (ibd)((inflammatory bowel disease) OR (ibd)) AND ((nsaid*) OR (non-steroidal anti-inflammatory drug*) OR (nonsteroidal anti-inflammatory drug*)) AND ((CRC) OR (colorectal cancer)) AND (meta-analysis[Filter] OR review[Filter] OR systematicreview[Filter])

Articles were selected based on being a systematic review, review, or meta-analysis. Further reading eliminated articles that did not meet the specified criteria. Additionally, articles were excluded from the search if they were not in English. Two independent authors screened the articles to ensure they met the necessary criteria [33].

## 3. Results

The Cochrane Library population also yielded 70 results for “‘gastroenterology and hepatology’ and ‘cancer: colorectal’“ and 91 for “’gastroenterology and hepatology’ and ‘inflammatory bowel disease’”. However, after reviewing the results, none of the articles discussed NSAIDs as chemoprevention for patients with IBD. For the searches on PUBMED, 18 articles were found for (“nsaid*”[All Fields] AND “ibd”[All Fields] AND “cancer*”[All Fields]), 9 for ((nsaid*) AND (colorectal cancer) AND (ibd)), 5 for ((aspirin) AND (colorectal cancer)) AND (ibd), and 101 hits. In total, 17 articles met the full selection criteria (Table 1).

Most of the published studies on this topic are case-reports or observational studies. There are few clinical trials. The conclusions from the review articles are summarized in Table 1. Among these, a 2009 article stated that there is inconclusive evidence regarding the chemopreventative properties of NSAIDs for CRC among patients with IBD [34]. Others have suggested that the long-term use of NSAIDs have potential protective effects against CRC for IBD patients, as most patients tolerate these medications despite the increased risk of unfavorable health outcomes, such as IBD flares, for some patients [35]. The dosage of the NSAIDs in these reviews is often not consistent or stated, reflecting the lack of information in the original sources. However, one review goes as far as simply not recommending any NSAID for chemoprevention for patients with IBD [36]. One anti-inflammatory drug, 5-aminosalycilic acid (5-ASA), is chemically similar to aspirin and showed promise as a chemopreventative agent among the general population, but the efficacy of 5-ASA’s chemopreventative properties remains largely debated [37,38]. Its mechanisms are not entirely known, but are thought to involve the inhibition of the cyclooxygenase and lipooxygenase pathways. Based on the literature searches, 13 of the 17 articles believed NSAIDs and anti-inflammatories, such as 5-ASA, have promising chemoprotective effects, but the others remain unconvinced of their use due to toxicity and the lack of research on the drug’s effects [30,39]. Two meta-analyses and one review concluded that exposure to 5-ASAs reduced the risk of colon neoplasia, dysplasia, and CRC. However, it is unknown if these events of dysplasia would progress to CRC [40,41,42]. Qui et al. determined that the chemopreventative effects were only significant at reducing the OR in clinical studies, not population-based studies. They also found chemoprevention with 5-ASA to only be significant for patients with UC [42]. This does raise questions on why there are discrepancies in the findings by study designs in the IBD population. Three of the searched papers concluded that a dosage of at least 1.2 mg/day of 5-ASAs are beneficial for CRC prevention, but the optimal dose for aspirin as a chemopreventative agent is unknown. The articles reviewed collectively show that there is still much unknown about the subject of CRC prevention for IBD patients with NSAIDs.

Eight of the 17 articles from the literature search note that the study design for NSAID trials among patients with IBD needs to be improved. The variability within epidemiologic designs is a challenge, particularly because most of the research currently looking into the relationship between NSAIDs, CRC, and IBD utilize a retrospective study design. In order to obtain a better understanding of the causative relationship, it seems important to use longitudinal studies, either randomized controlled trials or prospective cohort studies. However, conducting randomized control trials was criticized as a methodology in a few of the reviews due to the ethical implications of withholding 5-ASAs from patients, which are often taken for IBD to achieve remission, as defined by the relief of symptoms and a normally functioning immune system, and remission maintenance [43,44,45]. However, it should be noted that 5-ASA can still be administered rectally for intermittent flare-ups of IBD, whereas the potential chemopreventative effect of 5-ASA would be examined through regular intake at presumably low doses. Due to the commonality of an IBD diagnosis in adolescence and young adulthood, it is important to understand the longitudinal effects of a young adult or adolescent taking NSAIDs as a form of treatment for their IBD. A prospective cohort study can provide information on the longitudinal effects and methodologic issues such as compliance. A previous metaanalysis of prospective cohort studies assessing cancer risk with low-dose aspirin use found an overall protective effect for CRC among patients without an IBD diagnosis 0.76 (0.64–0.90) [44]. It would be beneficial to conduct these studies of low-dose NSAIDs for patients with IBD. Nevertheless, it would be helpful to better understand the biology of NSAIDs on CRC chemoprevention in IBD. First, the mechanisms of IBD and its progression into CRC are not known, and adding another mechanistic layer, presumably Cox-2 inhibition by NSAIDs, only leads to more uncertainty regarding the question at hand. Another consideration is that the mechanisms of action for NSAIDs might be different for CD and UC patients.

## 4. Discussion

The recommendation for a patient with IBD to take NSAIDs varies by provider. Many physicians do not recommend NSAIDs for patients with IBD due to the high risk of adverse effects; however, depending on the patient’s individual phase of the disease, it may not be of concern for the individual provider [45]. Mucoprotective drugs, such as rebamipide, could be taken with low-dose aspirin to minimize gastrointestinal toxicity, which could lead to a tailored CRC prevention strategy for patients with IBD [46]. Another future area of research involves using personalized medicine to determine which dose and type of NSAID is most effective for patients with IBD. Insights on the personalized NSAID type and dosage can include comorbidities, family history, gut microbiota composition, and other current medications. Using personalized medicine to determine optimal dosing and NSAID type based on each individual’s genotype could provide the most benefit to IBD patients due to the ambiguous nature of the genetic etiologies of the disease [47]. In order to conduct the above research ideas, the use of electronic medical records (EMRs) can provide quick and widespread data about the health history of patients with IBD. Two commonly used EMR databases are TriNetX and the IBM MarketScan Commercial Claims and Encounters Database. The IBM MarketScan Commercial Claims and Encounters Database includes insurance claims for inpatient, outpatient, and outpatient prescriptions, and TriNetX uses International Classification of Diseases (ICD) codes to track data [48,49]. By utilizing these databases, the data on IBD patients taking NSAIDs can be used to study their effects on developing polyps, CRC, and other cancers. However, the databases themselves are not complete. Such studies would also need to collect data on over-the-counter medications such as low-dose aspirin

Selective cyclooxygenase 2 (COX-2) inhibitors have potential as chemopreventative agents for patients with IBD. They are a class of NSAIDs that target the cyclooxygenase 2 pathway, which is often upregulated in cancer [50]. There has been backlash and hesitancy regarding the widespread use of COX-2 inhibitors due to links to cardiovascular toxicity, but this class of drug has shown great promise for reducing CRC risk. Two randomized controlled trials, the Adenomatous Polyp Prevention on Vioxx (APPROVe) trial and the Adenoma Prevention with Celecoxib [APC] study, showed significant CRC protective effects of rofecoxib and celecoxib, respectively [50]. However, it is important to note that rofecoxib (Vioxx) was pulled from the market as a result of severe cardiovascular adverse effects [51]. Celecoxib is still available and is used for many conditions and is recommended by the FDA as supplemental therapy for the prevention of colorectal polyps for patients with familial adenomatous polyposis [52]. Some researchers think the risk of cardiovascular impairment outweighs the benefits of the CRC chemopreventative properties of COX-2 inhibitors, while others believe otherwise. However, there are no viewpoints on this specifically for IBD patients. Therefore, it is important to conduct studies that can help minimize the risk of gastric and cardiovascular toxicity so the chemoprotective properties of COX-2 can be utilized. Taking celecoxib for a shortened timeframe of 12 months or less still led to chemoprotection of CRC, whereas other studies have observed patients taking celecoxib for up to 2.5 years [53]. Coupling celecoxib with cardiovascular intervention strategies has been suggested to mitigate serious cardiovascular effects, but its efficacy is unknown [54]. One meta-analysis containing studies with a five year follow up did show that coupling low-dose aspirin with celecoxib decreased the chemoprotective effects of celecoxib, but there was also a significant decrease in cardiovascular thromboembolic events [55]. Additionally, studies have shown inconclusive results on the impact of COX-2 inhibitors on IBD flare ups, so its use as a single agent for both chemoprevention and IBD treatment is unclear. It is important to continue researching ways to decrease the toxicity of selective Cox-2 inhibitors, particularly in IBD patients, to understand their potential role in CRC prevention for patients with IBD. It should be noted that Cox-1 has not traditionally been considered in CRC, although one study found increased expression in colorectal cancer tissue [56]. The Cox-1 signaling pathways have been implicated in intestinal polyp formation and a potential role of Cox-1 inhibition is a research avenue largely unexplored. One intriguing area is the effect of aspirin on platelet biology, which includes the inhibition of the release of cytokines and angiogenic factors by platelets [57]. The platelet count is increased in IBD, in addition to altered platelet function and biology, such as increased prothrombic cytokines. Cox-1-dependent prostaglandins control platelet activation. Thus, Cox-1 inhibition may have a role as a CRC chemopreventative mechanism in IBD [58].

A potential chemopreventative NSAID candidate for IBD is the nitric oxide-releasing (NO-releasing) NSAID. NO-releasing NSAIDs are standard NSAIDs with the addition of ONO2, which has been shown to induce oxidative stress and inhibit the downstream carcinogenic pathways in the colon. NO-releasing NSAIDs can still inhibit the COX pathway, preventing cell proliferation [59,60,61]. The question of the chemopreventative effects of NO-releasing NSAIDs has only been partially answered, as most of the studies on this topic have used in vitro and in vivo methods. The ability of NO-releasing NSAIDs to induce oxidative stress and modulate cancer progression has been proven in mouse models, thus proving their chemopreventative effects [62]. One study focused on the mechanisms of NO-ASA chemoprevention in mouse models and in human adenocarcinoma cells. In mice, NO-ASA was able to inhibit carcinogenesis without impacting healthy cell-proliferation, and in the human cells, NO-ASA induced oxidative stress, leading to the apoptosis of cancer cells [63]. The safety of NO-releasing NSAIDs has also been shown, as NO-releasing aspirin did not lead to gastrointestinal toxicity in mice and was even able to significantly reduce the toxicity of other NSAIDs when taken together [64,65]. However, safety data in humans is not yet available. NO-releasing aspirin is believed to be more potent and thus more effective at preventing CRC than regular aspirin, which is thought to be a standard property of the compound class [66]. Moving forward, it would be desirable to see studies comparing the safety and efficacy of various NO-releasing NSAIDs to determine which drug would be best for chemoprevention, without toxicity, in IBD patients. Although there has been great promise of NO-releasing NSAIDs in vivo and in vitro, it is still largely unknown how this class of drug will act in humans. A phase 1, double blind clinical trial to determine the safety and efficacy of NO-releasing ASA on colonic lesions in high-risk individuals was recently conducted, but there have not yet been any updates on the results [67]. The results from this study could provide beneficial insight on the potential of NO-releasing ASA in IBD patients.

Genetic studies can also be beneficial for determining the efficacy and safety of NSAIDs as a personalized chemopreventative agent among patients with IBD. It was found that the NOD2 and CARD15 genes can make individuals more susceptible to IBD development [8]. Another study found that individuals who use aspirin and other NSAIDS with the rs2965667-TT genotype have a 0.66 times lower risk of CRC [95% CI, 0.61–0.70], but those with the TA or AA genotypes have a 1.89 times higher risk of developing CRC. This study further describes that the regular use of aspirin and/or NSAIDS has a protective effect against CRC among those with the rs16973225-AA genotype (OR, 0.66 [95% CI, 0.62–0.71]) but not for those with the AC or CC genotype (OR, 0.97 [95% CI, 0.78–1.20]) [68]. The metabolism within individuals could impact the effect NSAIDs and aspirin have on the body, including the colon. UGT genetic variations were found to alter CRC risk. More specifically, certain polymorphisms, such as a haplotype in UGT2B15, were associated with an overall increased CRC risk (OR = 2.57, 95% CI = 1.21–5.04) and among the females (OR = 3.08, 95% CI = 1.08–8.74) in another study [69]. This is a nascent area with few studies, but presents many opportunities to examine how the effects of NSAIDs, as well as specific formulations, dosages, duration, medical condition may be modified by genetic variation. Gaining a glimpse of risk-altering variants within IBD patients can potentially guide clinicians to find effective and personalized chemoprotective agents.

The prolonged use of NSAIDs has been thought to lead to a myriad of health problems, and IBD has been part of this discussion. While this review focuses on NSAIDs as a potential chemopreventative agent in patients with IBD, it should be noted that NSAIDS might actually increase the risk of IBD itself. A cohort prospective cohort study in Europe supports the claim that aspirin use significantly increased the risk of CD (OR = 6.14, 95% CI = 1.76–21.35), but not UC, development (OR = 1.29, 95% CI = 0.67–2.46) [70]. Alternatively, a prospective cohort study of only women determined frequent NSAID use is associated with an increase in absolute incidence for IBD, but frequent aspirin use did not increase the absolute incidence of IBD. Although these results conflict with the European study, the results do draw on the importance of how one’s assigned gender impacts IBD development, that different formulations of NSAIDs may not have the same effects, and that determining a non-toxic dose for frequent NSAID intake is critical for considering its use as a chemopreventative agent [71]. In addition to these studies, there are several more, and a meta-analysis and systematic review found no significant association between NSAIDs and IBD risk and a lack of a plausible causative relationship [72]. However, the literature is incomplete with respect to several factors, such as the dosage of the medication and genetic factors. To address genetics, one in-vivo study that used mice concluded that NSAID use in IL-10 deficient mice led to the development of severe and chronic IBD [46]. By looking into specific genetic factors, researchers and clinicians could gain insight into whether patients with specific genotypes should avoid NSAID use altogether due to the likelihood of IBD onset.

## 5. Conclusions

The major gaps in the research suggest a strong need for more research on the relationship between NSAIDs and CRC prevention among patients with IBD to determine the lowest-effective dose of 5-ASA and aspirin that has a protective effect with the least toxicity. If the risk of IBD symptoms is minimized and the dose still reaches efficacy, the benefits of taking NSAIDs may outweigh the risks for patients with IBD. In contrast to the general population, there are opportunities to employ new research methodologies for IBD patients. For example, the American Gastroenterological Association [AGA] recommends screening for CRC 8–10 years after the first onset of IBD symptoms, and at subsequent intervals every 1–2 years. This frequent screening allows for prospective evaluation for the development of high-risk colonic polyps in relation to NSAIDs. Such designs also allow for clinical trial designs that are perhaps more conclusive than observational prospective studies. The study length of the follow-up time to adenoma development or a suitable inflammatory biomarker would be much less than for frank colonic carcinoma. Further, such studies can be conducted in the younger age groups that experience IBD. Generalizing the NSAID study results from older populations who use NSAIDs to these younger populations is uncertain, and direct findings in IBD patients who may have further autoimmune deficiencies and who take other IBD-related medications will be informative. Such studies should be designed to specifically collect medication doses.

## Figures and Tables

**Table 1 cancers-15-01102-t001:** Summary of Articles.

Author(s)	Title	Year	Conclusions
Eaden	Review article: the data supporting a role for aminosalicylates in the chemoprevention of colorectal cancer in patients with inflammatory bowel disease	2003	There is not enough data to conclude that 5-ASA is an effective chemopreventative agent for CRC in IBD patients.
Ryan et al.	Aminosalicylates and colorectal cancer in IBD: a not-so bitter pill to swallow	2003	5-ASAs provide protection against CRC and could be used in the general population and IBD patients
Cheng and Desreumaux	5-aminosalicylic acid is an attractive candidate agent for chemoprevention of colon cancer in patients with inflammatory bowel disease	2005	5-ASA is a safe and effecting chemopreventative agent for IBD patients, but more studies, preferably randomized control trials, are needed to further assess the use of 5-ASA.
Giannini et al.	5-ASA and colorectal cancer chemoprevention in inflammatory bowel disease: can we afford to wait for ‘best evidence’?	2005	Long term IBD treatment with 5-ASAs shows CRC prevention, and the use of it as a chemopreventative agent for IBD patients should be prioritized.
Van Staa et al.	5-Aminoalicylate use and colorectal cancer risk in inflammatory bowel disease: a large epidemiological study	2005	The consistent use of 5-ASAs over a long duration of time can reduce CRC risk in IBD patients.
Velayos et al.	Effect of 5-aminosalicylate use on colorectal cancer and dysplasia risk: a systematic review and metaanalysis of observational studies	2005	There is a protective relationship between 5-ASAs and CRC for patients with UC.
Chan and Lichtenstein	Chemoprevention: Risk Reduction with Medical Therapy of Inflammatory Bowel Disease	2006	5-ASA compounds have been well-studied and show a dose of 1.2 g/d is likely effective for CRC prevention in DC and UC.
Munkholm et al.	Prevention of colorectal cancer in inflammatory bowel disease: value of screening and 5-aminoalicylates	2006	The risk of colorectal neoplasia can be reduced with use of 5-ASAs, although dosing suggestions are not stated. There is limited information on prospective clinical trials.
Rubin et al.	Colorectal cancer prevention in inflammatory bowel disease and the role of 5-aminosalicylic acid: a clinical review and update	2008	There are still gaps in research surrounding 5-ASA as chemoprevention for patients with IBD, although there does appear to be promising results.
Droste et al.	Chemoprevention for colon cancer: New opportunities, fact or fiction?	2009	More studies need to investigate long-term use of 5-ASA for chemoprevention, as it is the primary maintenance medication. Selective COX-2 inhibitors could provide effective chemoprevention with minimal GI-toxicity.
Herfarth	The role of chemoprevention of colorectal cancer with 5-aminosalicylates in ulcerative colitis	2012	5-ASAs should not be used solely for chemoprevention of CRC in IBD patients
Margagnoni et al.	Critical review of the evidence on 5-aminosalicilate for chemoprevention of colorectal cancer in ulcerative colitis: a methodological question	2014	5-ASAs can decrease incidence of CRC and are an ideal chemopreventative candidate for IBD patients
O’Connor et al.	Mesalamine, but Not Sulfasalazine, Reduces the Risk of Colorectal Neoplasia in Patients with Inflammatory Bowel Disease: An Agent-specific Systematic Review and Meta-analysis	2015	When mesalamine is administered at greater than 1.2 g per day, there is a protective effect against the development of colorectal neoplasia.
Burr et al.	Does aspirin or non-aspirin non-steroidal anti-inflammatory drug use prevent colorectal cancer in inflammatory bowel disease?	2016	NA-NSAIDs and aspirin are mechanistically promising for the use of chemoprevention in IBD, but there is limited clinical evidence to make that conclusion.
Bonovas et al.	Systematic review with meta-analysis: use of 5 aminosalicyclates and risk of colorectal neoplasia in patients with inflammatory bowel disease	2017	5-ASAs have been shown to significantly reduce the risk of colorectal neoplasia, only in patients with UC.
Qui et al.	Chemopreventative effects of 5-aminosalicylic acid on inflammatory bowel disease-associated colorectal cancer and dysplasia: a systematic review with meta-analysis	2017	5-ASA is more effective at reducing the risk of CRC in patients with UC than CD. A dose greater than 1.2 g per day is most protective.
Abdalla et al.	Role of Using Nonsteroidal Anti-Inflammatory Drugs in Chemoprevention of Colon Cancer in Patients with Inflammatory Bowel Disease	2020	The long-term use of NSAIDs is necessary to provide a protective effect, but patients can be at risk for negative health outcomes.

## Data Availability

No new data were created or analyzed in this study. Data sharing is not applicable to this article.

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
