# Peer review of "Potential Role of Non-Steroidal Anti-Inflammatory Drugs in Colorectal Cancer Chemoprevention for Inflammatory Bowel Disease: An Umbrella Review"

_cancers, 2023, doi:10.3390/cancers15041102_

Round 1

Reviewer 1 Report

the topic of chemoprevention of colorectal cancer in IBD is an important  topic which is reviewed in this manuscript. I have the following comments

1- The authors mixed between NSAIDs and 5-ASA. 5-ASA is not considered a NSAIDs and is given for treatment of IBD. There are only two reviews on the use of aspirin and NSAIDS in the chemoprevention of CRC and all the other reviews are on 5-ASA. It should be made clear which class of drugs we are discussing. Actually, the title and the background discuss mostly the effect of NSAIDs, yet most of the reviews are about 5-ASA. The term 5-ASA was not used in the literature search. There are other reviews on 5-ASA in the literature if the aim of the authors is to review 5-ASA effect.  

2- the manuscript should be more focused on the subject and should be made shorter by 30-50%. For example, the lengthy discussion about prevalence and incidence of IBD in the introduction and figure 1 are not relevant and should be omitted. Instead, more discussion about CRC incidence and risk factors can be provided

3- it would be of interest to give some numerical figures about the magnitude of the chemoprotective effect of NSAIDs and 5-ASA  and the clinical relevance. Some data about the time needed to see the effect of these drugs is also of interest.

4-    lines 75-76:  'immunosuppressive therapy for IBD may increase the risk for CRC and other cancers'. this is true for some cancers but not for CRC

5-lines 95-96: this is not true. NSAIDs are not used to treat abdominal pain during flares

6- title: replace 'and' with 'in' 

Author Response

A summary of all edits can be found in the attached document

Reviewer 2 Report

This manuscript is a narrative review of the use of low-dose non-steroidal anti-inflammatory drugs (NSAIDs) for chemoprevention against colorectal cancer (CRC) in patients with inflammatory bowel disease (IBD). The modest scope is appropriate for a short review, and the topic is of current interest. However, there is a major conceptual gap in the discussion section and several other issues to be addressed as well.

Major issue: The authors mention only the COX-2 pathway connection with tumorigenesis (line 85) and neither COX-1 nor platelets are brought up later in the Discussion section. However, many observations link the COX-1 pathway with cancer risk. More immediately, the low-dose aspirin regimens being considered for chemoprevention resemble those widely used for cardioprotective effects, with the latter attributed to inhibition of platelet COX-1. Expanding the discussion to consider possible roles of COX-1 inhibition and anti-platelet effects in chemoprevention would make the review more informative and useful.

Other issues:

1) Insert "colorectal" before "cancer" in line 3 to reflect scope of review.

2) Thorough proofreading needed to fix typos and other glitches. A few examples: line 110 (intended meaning of "from the start of the paper"?); Table 1(chronological order more informative than by author name); Table 1 (ensure reference list has complete citations for all studies); line 171 (change "withholding patients from 5-ASAs" to "withholding 5-ASAs from patients"); line 205 (rephrase to clarify intended meaning of "Both databases pull data differently but can lead to similar results");  line 211 (intended meaning of "collect non-prescription data such as low-dose aspirin "?); lines 216-219 (citations for the cox studies?); line 239 ("in CRC prevention. with IBD ").

Author Response

(The authors gave the same response as above.)

Round 2

Reviewer 1 Report

the authors did not address the fact that 5-ASA is not a NSAID. Mixing them together is misleading.

the text is still too long and should be more focused.

Author Response

Added clarification for 5-ASAs starting at line 81. A copy of the statement can be found attached below. 
